# Design and Construction of a Portable IoT Station

**DOI:** 10.3390/s24134116

**Published:** 2024-06-25

**Authors:** Mario A. Trape, Ali Hellany, Syed K. H. Shah, Jamal Rizk, Mahmood Nagrial, Tosin Famakinwa

**Affiliations:** 1School of Engineering, Design and Built Environment, Western Sydney University, Sydney, NSW 2747, Australia; a.hellany@westernsydney.edu.au (A.H.); 17899534@student.westernsydney.edu.au (S.K.H.S.); m.nagrial@westernsydney.edu.au (M.N.); t.famakinwa@westernsydney.edu.au (T.F.); 2Urban Transformation Centre, Western Sydney University, Sydney, NSW 2747, Australia

**Keywords:** PLC, industrial IoT, modular IoT architecture, sensor data synchronization

## Abstract

This paper discusses the design and implementation of a portable IoT station. Communication and data synchronization issues in several installations are addressed here, making possible a detailed analysis of the entire system during its operation. The system operator requires a synchronized data stream, combining multiple communication protocols into one single time stamp. The hardware selected for the portable IoT station complies with the International Electrotechnical Commission (IEC) industrial standards. A short discussion regarding interface customization shows how easily the hardware can be modified so that it is integrated with almost any system. A programmable logic controller enables the Node-RED to be utilized. This open-source middleware defines operations for each global variable nominated in the Modbus register. Two applications are presented and discussed in this paper; each application has a distinct methodology utilized to publish and visualize the acquired data. The portable IoT station is highly customizable, consisting of a modular structure and providing the best platform for future research and development of dedicated algorithms. This paper also demonstrates how the portable IoT station can be implemented in systems where time-based data synchronization is essential while introducing a seamless implementation and operation.

## 1. Introduction

The implementation of sensors that consume minimal power and single-node data transfer using narrowband networks are widely used in IoT systems. These systems are developed and deployed through countless applications, including medical [1], global positioning systems [2], intrinsic safety systems [3], and light spectrum analysis systems [4]. Regardless of the application in which the IoT system is deployed, all systems have certain aspects in common, such as a self-powered sensor with an internal or external battery, a limited number of registers published into the network, and data refresh rates greater than 1 s; in addition, despite being interfaced into a single communication port, not all devices are synchronized with a real-time stamp. This affects any detailed analysis of the data acquired.

This paper aims to present a portable device using the Internet of Things (IoT), which combines industrial peripherals and sensors, providing a platform for the synchronization of systems used in general or specific applications, such as photovoltaic farms, wind turbines, hybrid renewable energy systems (HRESs), and general industrial applications. The structure, topology, and interface of the portable IoT station will be described, and simplified examples of data formatting and pre-processing, published in a single communication bus network managed by a programmable logic controller (PLC), will be provided. 

Currently, any portable instrumentation utilized in an HRES for troubleshooting or performance evaluation acquires individual readings, which are not compiled, processed, or published into a network. This characteristic influences the overall productivity of the technician or engineer, which assesses the system to identify an issue or to acquire systematic metrics for predictive or corrective maintenance. 

Depending on the occasion, a failure might occur in certain stages of the operation of the system that is being analyzed. For example, most failures in a photovoltaic installation occur when the solar irradiation or the temperature index is above certain levels, predefined by the system’s overall performance. When an individual rating is analyzed, the data synchronization and analysis might be inaccurate since the technician can only analyze each rating individually. 

The synchronization of multiple systems through a single communication bus is essential for system optimization and load management since every data register utilizing each device must be delivered simultaneously to avoid data mismatch. Several platforms can be created in the portable IoT station, and each platform consists of hardware blocks, which can be customized to a specific application, being system control, supervision, or integration.

The objective of this paper is to propose a centralized solution, operating as an edge computer capable of collecting multiple readings from decentralized sensors, controllers, and peripherals, eliminating any complications with data synchronization. Consequently, this allows the technician or engineer responsible for commissioning or troubleshooting the HRES to deploy this unit in any pre-existing system. This paper describes the overall performance of the portable IoT station by analyzing the response time of a variable from field acquisition to visualization utilizing an MQTT server and a locally hosted approach through a wired or wireless connection.

The methodology implemented in the portable IoT station follows edge computing principles. All variables and registers defined across a system are combined into a PLC, which is responsible for conditioning and publishing these variables and registers into a single Modbus transmission control protocol and internet protocol (TCP), open platform communications united architecture (OPC-UA), or message queuing telemetry transport (MQTT) server. 

The concept of having a portable IoT station to compile multiple communication protocols into a single communication interface utilized for troubleshooting and analysis of HRES has already been discussed [5]. This paper evolves the conceptual system into a prototype showing how the system can be designed, assembled, and programmed to achieve its desired outcome. All components utilized in the portable IoT station are classified as industrial grade by achieving and exceeding existing industry standards. By using these components, the system proposed in this paper is considered an industrial IoT system (IIoT) subsequently enabled to be deployed into any industrial application, including energy management and production. This paper incorporates the following sections:Research ObjectivesRelated WorkReview of Challenges in Portable IoT SystemsSystem StructureConsiderations for Cyber SecuritySystem TopologySystem ImplementationSystem CommissioningPerformance, Results, and System OptimizationFuture WorkConclusions

## 2. Research Objectives and Contributions

The following provides a summary of the objectives in this article:Develop a portable industrial Internet of Things (IIoT) system that can be integrated into a multi-platform environment, where all devices report their data into a unified time stamp.Employ middleware to coordinate and manage data across a decentralized network.Implement a programmable logic controller (PLC) as an edge computer to consolidate multiple data registers into a single communication protocol, specifically Modbus TCP.Implement a modular hardware structure, allowing an application-based customization of the system.

The following provides a summary of the contributions in this article:

Describe the hardware and system topology used in constructing the portable IoT station.Present two applications that illustrate the implementation and commissioning process of the portable IoT station.Provide an analysis of data latency from the sensor to the human–machine interface (HMI), as observed with the portable IoT station.

## 3. Related Work

As stated previously, there are various applications for portable IoT systems. Over the past few years, the need for weather stations propelled the implementation of sensors interconnected through a mesh topology network, forming a decentralized application of IoT systems. This topic was outlined by Novianty et al. [6]. The authors proposed a cost-effective solution supported by a prototype that enabled direct communication between the sensor and the MQTT server. A similar approach for a low-cost IoT system was implemented by Anik et al. [7]. However, the authors decided to integrate the sensors into Building Data Lite (BDL) instead of reporting the data through a mobile app or web server. Several other authors utilized low-cost IoT hardware to implement weather sensors into a decentralized network with microcontroller-based platforms, such as Arduino, Raspberry-PI, and NodeMCU [8,9,10]. Despite having accessible hardware based in a low-cost platform, each of these systems presents limitations on the product lifespan and data refresh rate. In 2023, Hachisuca et al. [11] developed a prototype combining customized hardware with sensors commonly used in weather stations to create an ideal solution for agricultural applications. The topology proposed used most of the potential of IoT systems in a decentralized application, connecting several nodes into a single IoT gateway and reporting the data to an MQTT server network. This topology was also utilized by Dubey et al. [12] in 2023. Dubey and associates aimed to create a new farming process based on the data collected across the farm, including the integration of the IoT system with irrigation systems. 

In both papers, Hachisuca et al. and Dubey et al. addressed the creation of flags and alarms that can trigger an additional process, correlating the IoT network with a supervisory system and ensuring that action is taken on the basis of the status of the farm. This approach was also utilized by Gsangaya et al. [13], notwithstanding the fact that having a decentralized network of sensors with self-power platforms counting on battery storage only introduces limitations to the overall autonomy of the system. Furthermore, to address the energy limitations of a decentralized IoT network, Warnakulasooriya et al. [14] developed a portable weather station with an IoT network coupled with a solar charger to power the internal battery bank, significantly increasing the autonomy of the system. 

The implementation of auxiliary power systems coupled with a portable or fixed IoT system was also implemented by Mariadass et al. [15] with a potable solar mobile system operating as a power distribution device, providing AC and DC energy supplies for general applications. Unlike other prototypes and topologies already cited in this paper, Mariadass et al. did not aim to have a decentralized IoT network; instead, they wanted to use the IoT network to report the status of their prototype which consisted of a battery cell, solar charger, solar panels, modular power supply, and an inverter. The same structure was utilized by Putra et al. [16,17]. These researchers utilized a redundant communication network with LoRaWAN and Wi-Fi to ensure that either communication protocol would reach the MQTT server.

A design factor that must be considered in all projects referenced as related work in this manuscript is the system’s resilience toward hardware and software structure. All designs comparable to the portable IoT station are based on low-cost systems with user-friendly interfaces and simple hardware integration, but they are not always reliable and robust against elevated temperature ratings and outdoor environments. Both factors can cause permanent damage to the system and generate unreliable data. Additionally, NodeMCU and Raspberry PI are both open-source and open-hardware devices, which enables data vulnerability due to direct access to operating system (OS) source codes and printed circuit boards (PCBs). Each system referenced as a relevant work uses certain aspects of the system topology and hardware structure implemented on the portable IoT station. Unfortunately, the system proposed in this paper does not have a direct comparison for its hardware structure, firmware configuration, or desired application. Nonetheless, the literature review allowed some of the challenges commonly identified in IoT systems to be highlighted and addressed on the portable IoT station.

Currently, there are no records of a portable IoT station that has configurable hardware, a programmable logic controller, an internal uninterruptable power supply, redundant communication protocols, and a multiport hardware interface. These features are a consequence of a new architecture compiling several communication protocols, enabling various components, sensors, and gateways to receive, process, and synchronize the data collected. Meanwhile, reporting or publishing it to a dedicated server is presented.

However, the internal communication network based on Modbus TCP is a common approach for IIoT and devices compatible with Industry 3.0 and 4.0 trends. Folgado et al. [18] summarized the current and future market trends for IIoT systems compliant with Industry 4.0. System digitalization, automation, and data-driven decision making are examples of features required for a contemporary industrial system. According to this analysis, features that are not present in the portable IoT system are included in the future work section of this manuscript. The system designed and implemented by F. Žemla et al. [19] successfully demonstrated the implementation of a digital twin with the utilization of Node-RED as a middleware and MQTT and OPC-UA server integration. This paper proposes a potential new market trend followed by Industry 5.0, which consists of all characteristics highlighted by Folgado et al. and includes augmented reality (AR) and mixed reality (MR) as requirements for a “Human-Centric Solution”. This paper sets an example of what can be accomplished in the future and shows how to achieve Industry 5.0. 

If the system developed by F. Zemla et al. were to be converted to the hardware structure utilized in the portable IoT station, the scientific and industrial contributions of such a device would be profound.

## 4. Review of Challenges in Portable IoT Systems

According to the latest publications on portable IoT systems, the challenges can be summarized as follows:Data flow and refresh rate;Energy autonomy;Redundant communication protocol.

### 4.1. Data Flow and Refresh Rate

To address the data flow and refresh rate issues on IoT systems with decentralized networks, the portable IoT station increases the data flow bandwidth to the user interface with alternative communication protocols, which include LTE and direct Wi-Fi connection. Several factors may impact the data flow bandwidth in a wireless network. In this paper, a comparison between wired and wireless networks was carried out based on the following: firstly, the amount of data submitted; and secondly, the time required for the system to be updated through the network.

In addition to the comparison conducted between wired and wireless networks, both systems had a data latency reduction in the overall processing time required by the PLC to receive and publish a global variable back to the gateway with the implementation of input filters with conditional functional blocks at the web interface of the PLC. These filters are configurable and can include basic logic functions. In both applications presented in this paper, three filters were applied, ensuring that only registers with variations above 5% of the previously registered value were transmitted to the rest of the network.

### 4.2. Energy Autonomy

To address energy autonomy, the portable IoT station has an uninterrupted power supply designed based on the IEEE Std 485-2020 [20]. Allowing the portable IoT station to be operational from 1.5 to 2 h depends on the number of inputs connected to the PLC interface. In addition, the station has an integrated power supply, allowing the station to be stationary.

### 4.3. Redundant Communication Protocol

There are in total eight communication protocols implemented into the portable IoT station: Wi-Fi, LoRaWAN, LTE, Modbus RTU, Modbus TCP, XanBus, Local Area Network (LAN), and Universal Serial Bus (USB). The data can also be published into a MQTT or OPC-UA server. In summary, there are at least two communication protocols available for every type of data transmission required in the portable IoT station:Wireless communication: Wi-Fi, LoRaWAN, and LTE.Wired communication: Modbus TCP, XanBus, and LAN.Serial communication: Modbus RTU and USB.

## 5. System Structure

The hardware proposed for the portable IoT station is specifically designed to synchronize all data registers existing in an HRES. Its structure targets three main functions: power quality analysis, troubleshooting, and system supervision. It is important to note that the proposed system is portable and needs to be self-powered. This feature enables the station to be transported to various locations and does not require additional infrastructure. This characteristic makes it possible for the portable IoT station to be connected directly to remote installations, for instance, small photovoltaic installations or small energy storage systems, where standalone applications are commonly applied without grid integration. 

The portable IoT station is not only ideal for HRES but also for any industrial application where remote diagnosis or supervision is required. In addition to the flexibility provided by the portable IoT station, it allows the data to be displayed in many forms and gathered through direct USB and Ethernet connection or wirelessly since all the data acquired by the station is published in an MQTT server via LTE or Wi-Fi connection. This process permits the system operator to visualize all data with a cell phone, laptop, or tablet through a web browser interface. 

The hardware used and the layout applied to the case enables the system to operate continuously in any environment with ambient temperatures varying from −10 °C to 45 °C. Using a PLC and an energy meter (EM) means that the system acquires voltage and current, and its fundamental characteristics, such as frequency, harmonic distortion, power factor, and load characteristics. 

The system integrated into the portable IoT station has a block structure classified into five independent systems. Figure 1 depicts the general arrangement of the internal assembly used on the portable IoT station; it also illustrates the location of each block structure. Figure 2 shows the location of each connection utilized under the external interface block. 

Table 1 documents the list of materials used in the portable IoT station, and this list can be used to replicate the prototype presented in this paper. 

### 5.1. Power Supply (PS)

The PS block allows the portable IoT station to have the UPS charged and to have the means to operate indefinitely. This is possible as long as an external energy source with a nominal voltage varying from 110 to 240 VAC is supplied and connected to the unit.

### 5.2. Uninterruptible Power Supply (UPS)

The UPS block enables the station to be energized without an external energy source, allowing the unit to be operational during transport. The topology applied to this block is essential to ensure that the unit can be fully portable, allowing up to 2 h of autonomy based on the installed battery module of 3400 mAh. 

The UPS control has an inbuilt DC/DC converter, which ensures stability at the 24 VDC line until the battery module reaches its limits based on the State of Charger (SoC) pre-defined inside the UPS control module. To achieve the optimal longevity of the battery module, the UPS control is configured to maintain a charge rate of 0.4 C. Therefore, it is required to have the system connected to the main supply for at least 2.5 h to have the battery module fully charged.

### 5.3. Energy Meter (EM)

The EM block allows the data acquisition of the electrical characteristics of any system connected through the external interface depicted in Figure 3. This block is essential for troubleshooting and commissioning since it allows the operator to analyze the electrical system. The hardware was adapted to have external overcurrent protection at every connection point and calibrated with a current transformer calibrated to a ratio of 100:1 ampere. The implementation of this block is executed by a meter directly connected to the desired section of the system that is to be analyzed.

The hardware adaptation undertaken on the EM block enables the portable IoT system to connect to a single or three-phase system, with line-to-line voltages up to 440 volts AC and line currents up to 500 amperes. 

Macros pre-configured into the energy meter can detect total harmonic distortion (THD), individual harmonic distortion of voltage, and current up to the 63rd harmonic, line frequency, power factor, load characteristics, active, reactive, apparent power, phase sequence, and any fault related to the overall power quality of the system that is analyzed. 

Inside the firmware of the energy meter, an algorithm is responsible for load predictability since the unit stores the occurrence of peak loads and attempts to map patterns with the date and time of those events. To communicate the data from the EM to the rest of the portable IoT station, a Modbus TCP connection is utilized. There are dedicated registers to access the data logged into the unit, and according to that, there is no need for additional filtering or conditional code lines on the data exported by the EM 220. 

### 5.4. Programmable Logic Controller (PLC)

The PLC block controls, converts, and manages all the other devices inside the IoT station. This block addresses dataflow and refresh rate issues on the portable IoT station, with an internal filter applied to its registers combined with an edge computing topology, combining all registers from the network into a single communication protocol. 

All functions on the PLC are adjusted and customized depending on the level of integration desired. The PLC utilized in this project has configurable modules that can be attached to the system, enabling digital and analog inputs and outputs. It also has a serial module to interface with any device using the serial recommended standard (RS) 485 or RS232. 

Typically, catastrophic fault conditions such as ground faults, grid islanding, system overcurrent, overvoltage, and undervoltage will force the power conversion unit (PCU) to send a signal with an external relay contact, indicating that a fault has occurred. The PLC configuration shown in this paper allows a total of 16 analog inputs, four digital inputs, and four dedicated inputs for the resistance temperature detector (RTD).

In addition to the hardware customization, the PLC can communicate directly to an OPC-UA and MQTT server via an Ethernet connection. This enables the publication of its global variables into a supervisory control and data acquisition (SCADA) system or allows its variables to be locally published to a human–machine interface (HMI). Referring to the portable IoT station, the PLC is connected via Modbus TCP to the rest of the system. It uses 4G LTE internet to publish its global variables to an MQTT server. The 4G network is generated by a modem that establishes a direct connection to the MQTT server. 

### 5.5. Additional Communication (ACOM)

The ACOM block allows the portable IoT station to initiate a direct interface with several communication protocols, enabling redundant communication lines to be implemented. This block is composed of gateways integrating several communication protocols into a single Modbus TCP network. One of the gateways is dedicated to the XanBus protocol, which is a communication protocol deployed by Schneider Electric. It serves in portable IoT stations to interface power conversion units (PCU), including inverters, rectifiers, battery management systems (BMSs), and transformers. 

The gateway used to establish XanBus communication uses Modbus TCP to communicate with the rest of the portable IoT station network. The system layout allows any gateway to interact with the portable IoT station if a Modbus port is available in the gateway. To complement the network interface of the portable IoT station, a low-power, wide-area networking protocol built on LoRa radio communication (LoRaWAN) gateway developed by Wattsense was incorporated into the system, converting the data from the LoRaWAN band to Modbus TCP. 

### 5.6. External Interface

To integrate the portable IoT station with external systems, a connection interface was developed. It assists the system operator with direct access to all ports required for the system integration, including USB, Ethernet, and four-wire direct ports.

### 5.7. Hardware Customization

As mentioned previously, most of the hardware utilized in the portable IoT station is commercially available. However, the construction of the portable IoT station required adaptations to convert a stationary din rail-mounted system into a portable interface. For that, structural changes had to be made on the casing to install a din rail where the components could be mounted, while a cable management structure was implemented following local wiring standards, such as AS/NZS 3000:2018 [21]. 

Once the din rail structure was installed, a 3D-printed conversion plate was required for the EM220 since the front surface of the casing is not flat. Additionally, surface-mounted fuses and “bullet” plugs were installed to convert the leads required for the EM220 to operate. All other connectors and plugs used in the portable IoT station had to be converted to surface mount structures and installed directly into the casing. For installation and temperature rise calculations, the AS/NZS 61439.1:2016 [22] was utilized. 

## 6. Considerations for Cybersecurity

All communication devices used in the portable IoT station comply with the following standards:IEC 62443-1-1:2009 [23]—Industrial communication networks—Network and system security, Part 1-1: Terminology, concepts, and models.IEC 62443-2-1:2010 [24]—Industrial communication networks—Network and system security, Part 2-1: Establishing an industrial automation and control system security program.IEC 62443-3:2013 [25]—Industrial communication networks—Network and system security, Part 3-3: System security requirements and security levels.

Each of these standards specifies encrypted communication protocols and local host passwords, ensuring restricted access to the system no matter which communication bus is utilized. Figure 4 illustrates the authorization process required by each communication protocol.

All connection access through USB, LAN, Wi-Fi, and MQTT requires a predefined user and password, which is encrypted and stored in the target device. However, Modbus TCP, Modbus RTU, XanBus, and LoRaWAN use end-to-end cryptography. LTE uses data packages following the Ciphering Key (CK). 

This paper did not evaluate the resilience of each communication protocol against cyber-attacks but ensured that all communication protocols in the portable IoT station followed the latest cybersecurity standards and guidelines provided by the organization which withholds intellectual property used in each communication protocol. 

Considering that the portable IoT station uses Modbus TCP as its internal network, it is important to emphasize that it currently complies with the “MB-TCP-Security-V21-2018” guidelines [26]. 

## 7. System Topology

As stated previously, the objective of the portable IoT station is to combine multiple systems into a single communication bus following the principles of an edge computer. The system topology might vary depending on, firstly, the objective of the portable station and, secondly, the equipment required to connect to the unit.

Figure 5 shows the system topology of the portable IoT station. In this topology, there are five systems in a single communication bus. Each variable addressed to a single register is converted from an Integer or Float point 32-bit hexadecimal defined by the IEEE 754-2019 [27] Floating-Point Arithmetic protocol. It is published to a Modbus TCP server host and converted to a decimal value in the PLC interface. 

Once the data gathered from the PLC through Modbus TCP is filtered, stored, scaled, and processed, it is published as a global variable in a configurable Modbus server. Subsequently, it is collected by the Wattsense server, published to an MQTT server through the 4G modem, and displayed in a web browser interface. The web browser interface from Wattsense allows all registers to be displayed, published, and organized with real-time acquisition and dynamic interaction. 

The usage of the PLC to centralize all registers available in the portable IoT station follows the principles of an edge computer. However, this approach can overload the hardware and firmware used on the PLC, resulting in long refresh rates. To address any system overload, the registers following a direct integer structure can bypass the edge device since this type of register can be published directly into the MQTT server if no additional filter is required. 

## 8. System Implementation

During the implementation of the portable IoT system, all Modbus RTU nodes using RS485 were established under 19,200 bps and even parity. All slave IDs for Modbus TCP and RTU were addressed from 1 to 247. Every register pulled and published in the Modbus TCP had the following structure: Hold registers (4XXXX) were set up as float and integer values, with read and write functions.Input registers (3XXXX) were all set up as float values with read-only functions.Input statuses (1XXXX) were set up as binary values with read-only functions.Coil statuses (0XXXX) were set up as binary values with read and write functions.

Table 2 summarizes the main components used in the portable IoT station; it also provides the address list utilized in the Modbus TCP network. 

Since the energy meter implemented into the portable IoT station accepts direct analog inputs through voltage leads and current transformers, these values are converted and allocated to certain registers within the network. Having the ability to host a Modbus RTU node allows the energy meter (EM220) to compile multiple registers to the Modbus TCP network. Additional features, such as digital inputs and output ports, allow many possibilities for condition monitoring and system supervision. The diagram on Figure 6 illustrates the internal data process flow used in the PLC collected from the EM220.

The utilization of the Schneider gateway, also identified by the manufacturer as the InsightHome gateway (865-0330), also allows additional registers relevant to the troubleshooting or commissioning of a system; with the utilization of XanBus, Modbus TCP, and Modbus RTU, it can collect fault codes, energy consumption and live status of the equipment connected. This unit also provides a web interface through a LAN or Wi-Fi connection. The advantage of XanBus over Modbus is the compatibility of the system with Schneider devices. All products manufactured by Schneider have a directory loaded into the gateway, so all the registers are promptly available in the gateway. 

The PLC selected for the portable IoT station is the UR20-WL2000-IoT. This device has a Node-RED interface for implementing several pallets promptly available from an online database as well as a native PLC language of function block and direct MQTT and OPC-UA server interface. Furthermore, it can host a Modbus TCP server to publish its global variables. 

All features required during the design, construction, and implementation of the portable IoT station are not exclusive to the UR20-WL2000; there are other models of programmable logic controllers that can be utilized as long as the controller has a native middleware interface or an Integrated Development Environment (IDE) such as CODESYS, which has multiplatform integration of several devices allowing an edge computing topology to be integrated into the PLC.

Using register four from the EM220 by requesting the register address 30,005 requires a data conversion in Node-RED, which is a browser-based flow editing system developed and optimized for IoT applications. Node-Red is considered a middleware since it enables the platform integration of different devices through a decentralized network and performs data processing of registers acquired with its native event-driven architecture. Each flow used in Node-RED is structured in pallets and nodes written in JavaScript.

The node utilized for the data acquisition is “Modbus Read”, which is then connected to a “toFloat” node. Once the data is converted, a “msg.payload” and “iodata-out” node converts the data value to a global variable and displays the status of the data into a debug interface.

Once the register is published as a global variable, it can be adjusted, scaled, and calibrated with the built-in function block interface installed in the PLC. Additional processes are required to convert the global variable into an integer with post-processing, and it is then exported directly to an OPC-UA and MQTT server or Modbus TCP network. 

The biggest advantage of using the UR20 is its modular capability. In this application, six analog cards, one RTD card, one output relay card, one serial card, and one digital output card were used. There are several cards available for this PLC. However, since the portable IoT station is optimized for power quality applications and system supervisory, having several analog inputs allows multiple sensors to be connected to the PLC simultaneously.

The Wattsense gateway served the purpose of data visualization since it uses a web interface accessible through any device. All the registers collected, compiled, processed, and published into the Modbus TCP network are accessible through the Wattsense tower. Despite the integration capabilities of the Wattsense tower, the interface does not make any data processing possible. Hence, there is a need for a PLC configured as an edge computer, filtering, processing, and publishing all the registers into a single Modbus TCP network. 

## 9. System Commissioning 

To simulate the commissioning process of the portable IoT station, two applications were created. Application “A” used remote data visualization, while Application “B” focused on local data visualization; the comparison between each application shows the data band with limitations based on the refresh rate of each topology used. This approach evaluates a technical term referenced by Sensor to Human–Machine Interface (S2HMI), where the sensor is the device or gadget collecting the data from the target application, and the Human–Machine Interface refers to the user interface referenced by a personal computer or a smartphone. 

Application A relies strictly on remote publication of the data acquired, using the 4G connection native on the Wattsense Tower to connect the system directly into an MQTT server, integrated into a web server, allowing the data to be visualized through a user interface. Application B uses a local host approach with the Wi-Fi network and Modbus TCP interface native to the Schneider gateway and PLC. In this application, a local LAN and USB network is also deployed, interfacing the connection directly into the PLC, with its native visualization tool based on Node-RED.

### 9.1. Application A

This application required a Fest Didactic product line with three different LabVolt modules. The PLC located inside the portable IoT station is responsible for communicating with the AC motor control module (8960-2A) and monitoring the system’s power quality with the EM220, which is connected to the source module (8821-2A). The power line is connected to a set of three-phase AC motors coupled to the module (8221-0A). The diagram below represents the overall system setup.

For this application, the following registers were defined and compiled into a single Modbus TCP address, with the utilization of Node-RED as a middleware managing the dataflow between the PLC and the rest of the Modbus TCP network following the same structure as depicted in Figure 7.

The system topology utilized during Application A can be visualized on Figure 8, which shows the LabVolt setup connected to the portable IoT station. Figure 9 shows how the test set up was configured. 

The flow charge shown on Figure 10 describes the process integrated into the PLC. External registers received through the Modbus TCP network are published into the Wattsense Console for data visualization. Additional steps can be added to integrate supervisory processes, such as actuator control or dedicated sensor visualization. 

Despite having the data published into the Wattsense Console for visualization, all the registers are put into the Node-RED visualization tool. This assists the user in troubleshooting and verifying the system time stamp.

Figure 11 shows the following user interface created on the Wattsense console with the web interface integrated into the MQTT server. 

In this application, all critical measurement and control features required to perform power quality analysis and load management were integrated with the EM220 and UR20. All data were processed and conditioned to be perfectly displayed through the Wattsense Dashboard. 

The biggest advantage of the methodology used in this application is the wireless capability enabled by the Wattsense gateway. It only requires a 4G (LTE) connection to establish a network between the portable IoT station and the MQTT server. In this way, any device to access the data acquired by the portable IoT station is allowed, without the need for proximity to the system. However, this approach results in delays in the data visualization. 

### 9.2. Application B

This application was executed with a simple HRES, which combined a solar installation, batteries, and a wind turbine at the same grid connection. The Schneider gateway (865-0330) was utilized. A hybrid inverter (Conext XW Inverter/Charger) is responsible for controlling the power output of the solar panels and operating the battery bank. The inverter was connected in parallel to the wind turbine controller. 

The wind turbine controller used in the system did not have external communication. To address this issue, the EM220 was connected directly to the wind turbine output since the Conext XW could monitor and report the power output of the grid connection. 

The communication between the Schneider gateway and the Conext XW Inverter/Charger was established through XanBus. Consequently, all the registers from each device were automatically addressed and integrated. Figure 12 illustrates the system topology used during Application B.

The same registers listed in Table 3, Table 4 and Table 5 were used from the EM220, UR20 and InsightHome gateway. Once again, Node-RED was implemented, managing the data flow between the PLC and the Modbus TCP network. 

The setup utilized during Application B can be visualized on Figure 13, which shows the Conext XW hybrid inverter connected to the wind turbine controller and a bank of batteries consisting of five Invicta LiFePo4 48V 75Ah through Modbus RTU. The EM220 was connected to the junction box of the wind turbine, while the Conext XW was connected to the InsightHome gateway through a direct XanBus connection. Through this process, all devices could be visualized on the portable IoT Station. 

The process flow displayed on Figure 14 shows how the registers acquired from the inverter through the XanBus communication, which is transmitted by the Schneider gateway through Modbus TCP into the PLC. Both visualization platforms are shown on Figure 15 and Figure 16.

This application aims to demonstrate additional visualization tools available through the portable IoT station. The data approach utilized for the data publication through Modbus TCP followed the edge computing topology, which ensures the required performance and data bandwidth. 

The biggest advantage of this approach is the refresh rate of the system; despite the number of variables acquired, this approach has a much better performance when compared to a remote methodology using a 4G (LTE) network, through Wattsense. This approach employed the portable IoT station as a local host, enabling the direct interface of the PLC and Schneider gateway into two distinct visualization tools. The data visualized had a high refresh rate since it had a minimum restriction on the data band. However, it required either a hardwired connection to the portable IoT station or close proximity to the Schneider gateway so that the Wi-Fi connection between the gateway and the computer was enabled. 

## 10. Performance, Results, and System Optimization

To analyze the performance of the portable IoT station, the overall refresh rate of the system was acquired. With this metric, the latency of the system was defined. This measurement served as the baseline for the optimization approach adopted in the PLC. 

To examine the refresh rate, a set value of registers was pulled from the Node-RED interface, acquiring the desired register from the Modbus TCP network. The debug function on Node-Red served to define the “start point”. To provide an accurate performance indication of the communication system, each test was repeated three times. To remove any additional variable, once a test scenario was acquired, a power cycle process would commence, resetting any existing buffer and cache memory that might impact the final results. The value displayed under the data latency evaluation is an average result of each repeated test. 

The latency response is based on the standard size of a Modbus register of 16 bits. The direct connection with LAN and USB was not evaluated; unfortunately, no baseline could be set for these communication lines. This is because the data transmission matched the data visualization used on the debug function of Node-RED. 

Additional registers were pulled from the Modbus TCP, such as the date, time, and operational hours logged on each device, allocated under Coil and Status registers. This approach allowed the system to be evaluated with up to 500 registers pulled from the network, increasing the data density of the system and showing its impact on the data rate. 

Figure 17 illustrates the performance evaluation used in this manuscript for the portable IoT station. The debug command is used to force the system to pull registers from the Modbus TCP network, starting the process flow shown in Figure 10 and Figure 14. The estimated time between the “start point” and the data visualization in each interface is used to define the overall data rate. 

The following Table 6 and Table 7 shows the performance acquired from Application A and Application B. All results are summarized into a single graph, which outlines the estimated data rate measured in bits per second and display against the number of registers pulled from the Modbus TCP network. 

### 10.1. Results Discussion

It is relevant to mention that any data transferred from the Node-RED interface to the user interface under one second cannot be recorded since the debugger requires at least one second to refresh. Considering this information, both applications recorded the same refresh rate when 10 registers were pulled from the Modbus TCP network. 

The refresh rate was defined as a period between the debug command on Node-RED, and the updated value at the targeted interface. Both applications meet the expected performance requirements. Application B achieved a much higher data rate than Application A, as shown in Figure 18. However, it is noted that the data rate from Application A is sufficient for most IoT applications. 

Both applications require at least a few seconds to update their user interfaces. This latency can cause the loss of several registers if the communication between the PLM and the target communication protocol is interrupted. 

### 10.2. System Optimization

To minimize issues resulting from the data latency existing in the portable IoT station, a simple optimization technique can be applied to the PLC by using an internal secure digital (SD) card available in the PLC as a buffer, where the register pulled from the Modbus network can be stored without any filtering or pre-processing. 

This ensures the instantaneous backup of the data received in the PLC, allowing the user to undertake further analysis of the data received if an error or failure occurs in the system within a period of 1 s. This technique was utilized by Novianty et al. [6] once the authors identified the latency of their system and the potential data loss caused by a lost connection between the gateway and the user interface. 

Unfortunately, the PLC does not have the capability of transferring the raw data from the Modbus TCP network directly to the SD through the IDE interface. Therefore, a dedicated flow from Node-RED must be implemented to collect all registers from the network and convert these registers into a .csv file with the time stamp collected. Once the .csv file is created, it can then be saved into the SD card. 

Incorporating a buffer into the PLC mitigates the impact of data latency on the network. However, this action also heightens the risk of a potential data breach, as a physical safety lock preventing unauthorized personnel from accessing the SD card within the PLC is lacking. While implementing a node credential in Node-RED could restrict access to the data stored in the .csv file, it fails to adhere to any industry standard regarding cyber security.

## 11. Future Work

### 11.1. Algorithm Implementation

Additional algorithms can be added to the portable IoT station, allowing further investigations and performance evaluation of any industrial system with data-driven decision making. Both applications presented in this paper are limited and do not explore the full potential of the system.

### 11.2. Large Scale Implementation

This system must be connected to a large-scale battery or solar farm to highlight its benefits during troubleshooting and performance acquisition. If this system is implemented in a large-scale installation, allowing more than 500 registers to be pulled from the Modbus TCP network will likely lead to a review of further issues in latency and data storage.

### 11.3. Energy Autonomy

Additional autonomy is required in the portable IoT station. During the system’s evaluation, it was identified that two hours of autonomy is not enough since the starting time requires at least 5 to 10 min to establish a connection across all the nodes. However, it is not possible to implement a bigger battery module since the internal space on the case is limited. The best solution is to include a solar charger following a topology similar to that of the portable solar station built by Putra et al. [17].

### 11.4. Industry 4.0

Currently, the portable IoT station is considered an IIoT system since it focuses on industrial applications and complies with all relevant industry standards. However, it does not comply with the latest industry trends defined by Industry 4.0. This statement is supported by the analysis performed by Folgado et al. [18] since the system presented in this paper lacks a digital twin implementation impacting its digitalization capability. This feature can be implemented with ThingWorx, which is an IIoT platform dedicated to the visualization and digitalization of IoT systems. ThingWorx can be implemented with the existing hardware and topology presented on the portable IoT station since the connection between the PLC and KEPServerEX, where the registers utilized on ThingWoxrs are located, is established by OPC-UA.

Before adopting Wattsense as the web interface for the portable IoT station, the conceptualization team responsible for the station attempted to integrate ThingWorx. However, ThingWorx necessitates a local server host to receive data from the portable IoT station. As a result, implementing this server into the Western Sydney University Network restricted access to the gateway installed on the portable IoT station, blocking the data transferred from the portable IoT station to the KEPServerEX. If this issue is addressed, a digital twin of the portable IoT station can be generated.

The hardware utilized in the portable IoT station complies with Industry 5.0 requirements. However, to achieve Mixed and Augmented Reality with gadgets such as MS HoloLens, the interface between the OPC-UA or MQTT server network must be implemented with KEPServerEX and ThingWorx, as mentioned as a limitation to achieve Industry 4.0. 

### 11.5. Cyber Security Assessment 

Although of all devices installed in the portable IoT station adhere to the latest cyber security standards, a vulnerability assessment can be performed to analyze the structure used on each device and web interface access. A similar approach to the one applied by Ramires et al. [28] can be performed, where a strategic offensive attack targeting register modification in a Modbus TCP network is applied; this approach will likely define vulnerabilities in the system. 

## 12. Conclusions

The portable IoT station will serve as a platform for data acquisition and synchronization, configured as an edge device. Its unique approach allows the station to be used as a tool to acquire and investigate the overall performance of the system connected. Since the portable IoT station is customizable, adjustments can be made to integrate new communication protocols and additional devices to perform specific tasks using Node-RED. 

The IoT station can operate as a supervisory tool with multiple analog and digital inputs using the existing hardware. Furthermore, it can serve as a control unit capable of connecting and disconnecting systems from the electrical network. For this reason, it is an ideal solution for researching and developing new algorithms for power quality analysis and load management. 

Despite the advantages of using the portable IoT station as a system integrating tool, there are limitations on the processing and publishing time required for each register. These limitations are caused by the data rate achieved between the Modbus TCP network and user visualization through a webserver using LTE, Wi-Fi, or even Modbus TCP. On this basis, the portable IoT station should not be implemented in industrial systems that require a prompt response with a time stamp lower than 1 s. Applications installed in explosive atmospheres constitute an example of the system where the portable IoT station should not be implemented, despite having most of its components certified for that type of application. 

The methodology utilized for the design and construction of the portable IoT station covers all technical challenges previously identified in portable IoT systems. Additional considerations were taken for component selection during the design stage, ensuring that the unit constructed is resilient under extreme weather conditions and against cyber attacks. 

The following outlines future academic implications stemming from the portable IoT station:Time base applications with apply and observe characteristics; as all data communicated within the network of the portable IoT station is synchronized, the margin allowed for error due to time shift is reduced.Seamless implementation and operation. The portable IoT station can be implemented into any system through several communication protocols. None of the protocols require hardware alterations on the target system, enabling a “plug and play” approach preferred by many academic institutions.

## Figures and Tables

**Figure 1 sensors-24-04116-f001:**
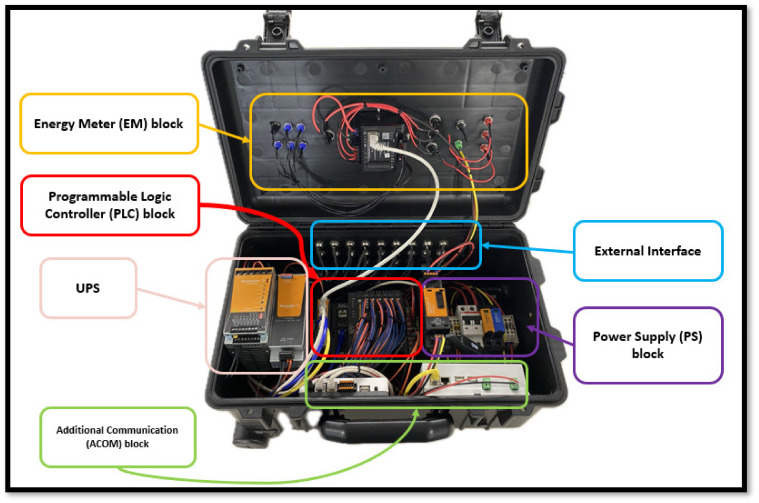
Block structure of the portable IoT station.

**Figure 2 sensors-24-04116-f002:**
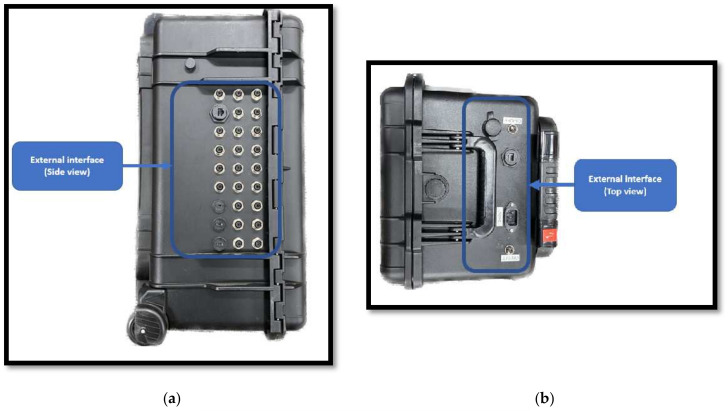
(**a**) The side view of the external interface of the portable IoT station; (**b**) the top view of the external interface of the portable IoT station; (**c**) the front view of the external interface of the portable IoT station.

**Figure 3 sensors-24-04116-f003:**
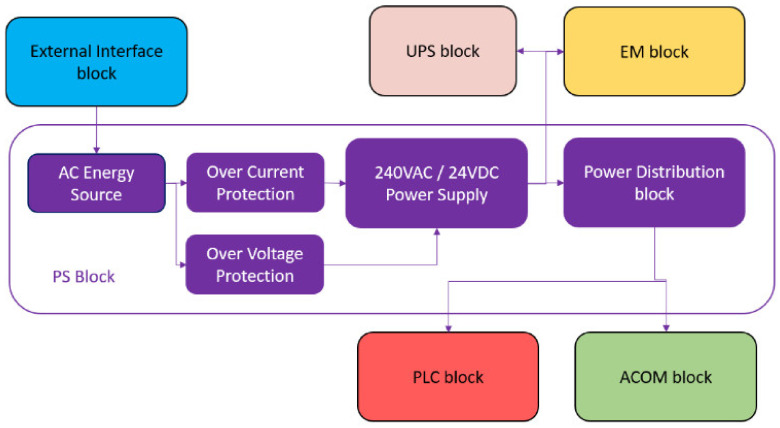
Simplified representation of the hardware integration between PS Block with ACOM, PLC, EM, UPS, and external interface block.

**Figure 4 sensors-24-04116-f004:**
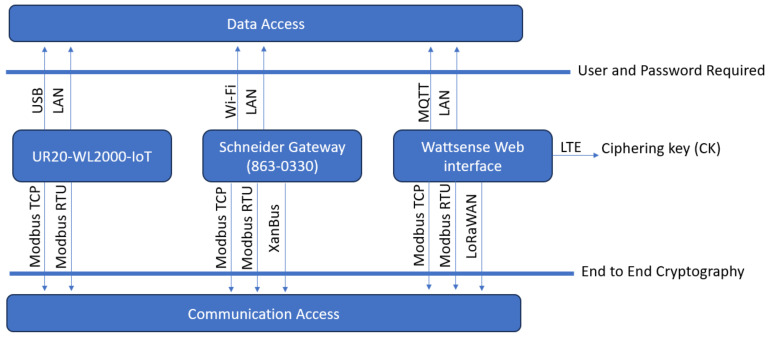
Authorization process required by each communication protocol.

**Figure 5 sensors-24-04116-f005:**
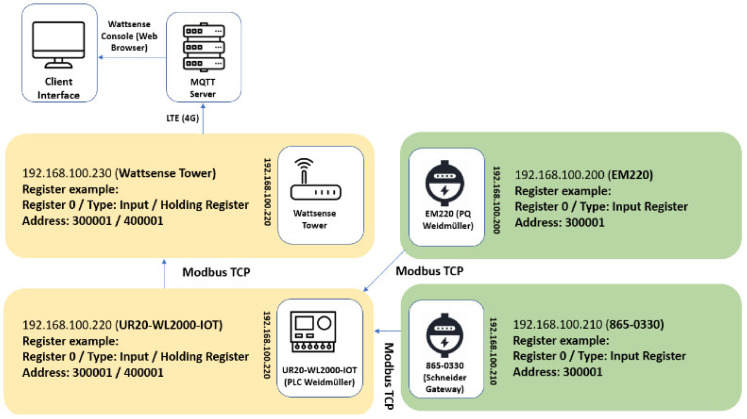
System communication topology.

**Figure 6 sensors-24-04116-f006:**
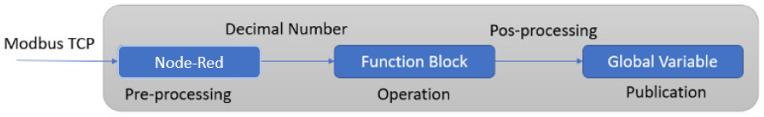
Internal data process flow used in the PLC.

**Figure 7 sensors-24-04116-f007:**
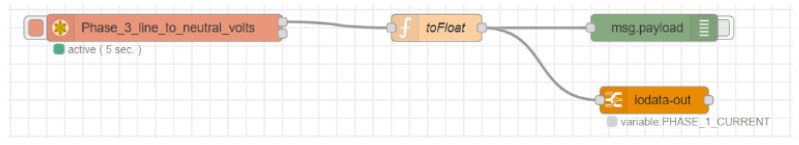
Node-RED dataflow.

**Figure 8 sensors-24-04116-f008:**
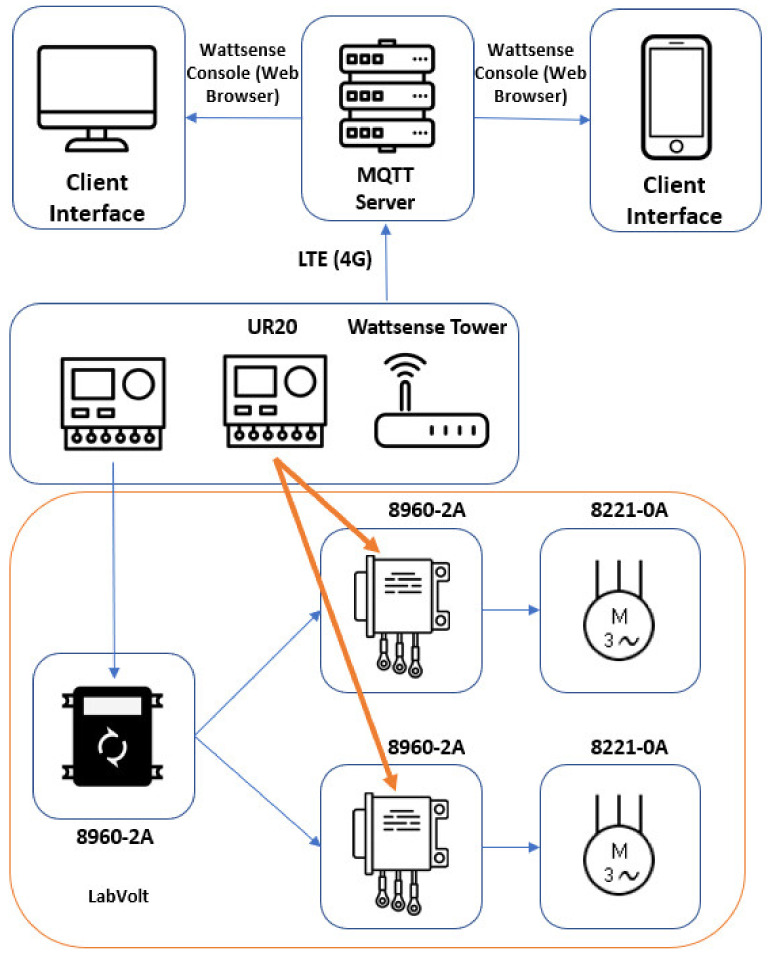
System topology applied during Application A.

**Figure 9 sensors-24-04116-f009:**
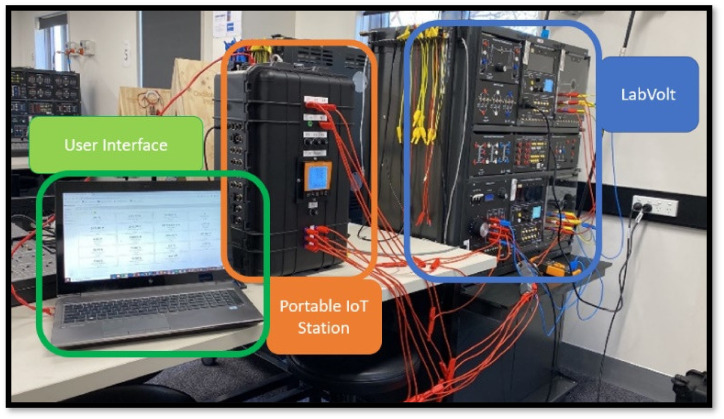
System configuration utilized during Application A.

**Figure 10 sensors-24-04116-f010:**
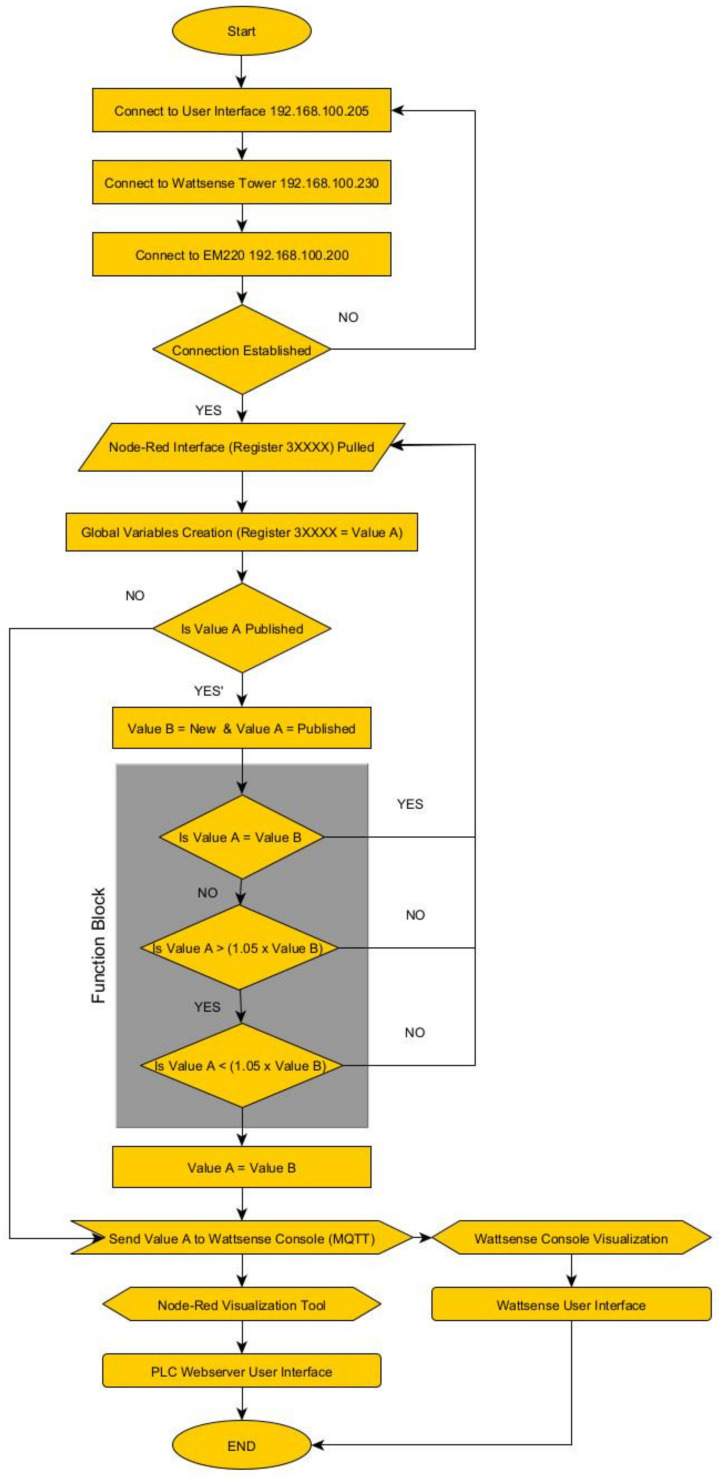
Process flow configured during Application A.

**Figure 11 sensors-24-04116-f011:**
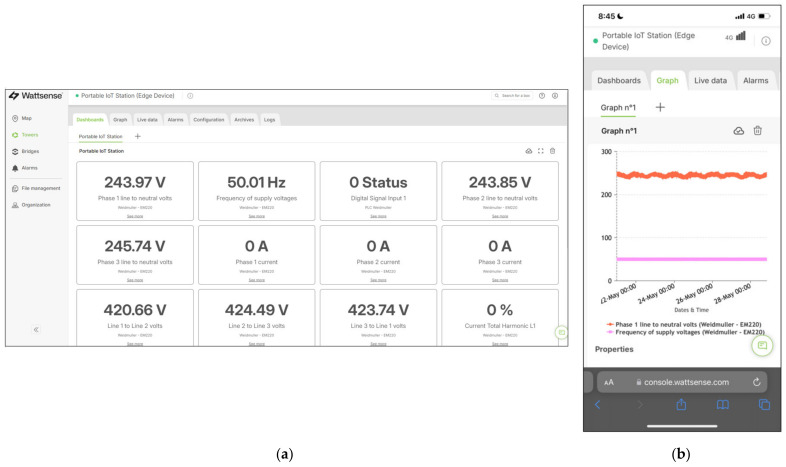
(**a**) The Wattsense console accessed through a computer; (**b**) the Wattsense console accessed through a smartphone.

**Figure 12 sensors-24-04116-f012:**
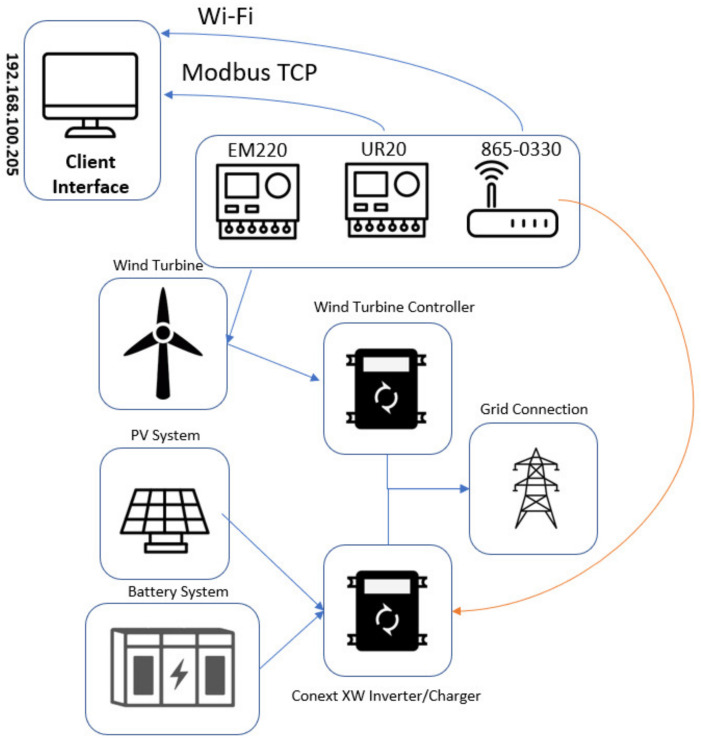
System topology applied during Application B.

**Figure 13 sensors-24-04116-f013:**
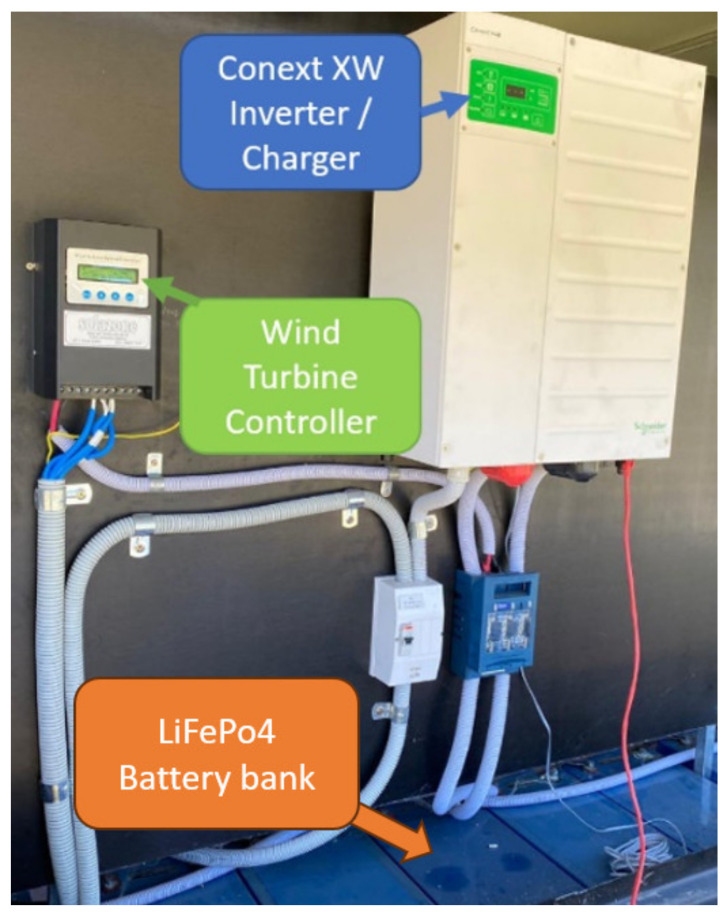
System configuration utilized during Application B.

**Figure 14 sensors-24-04116-f014:**
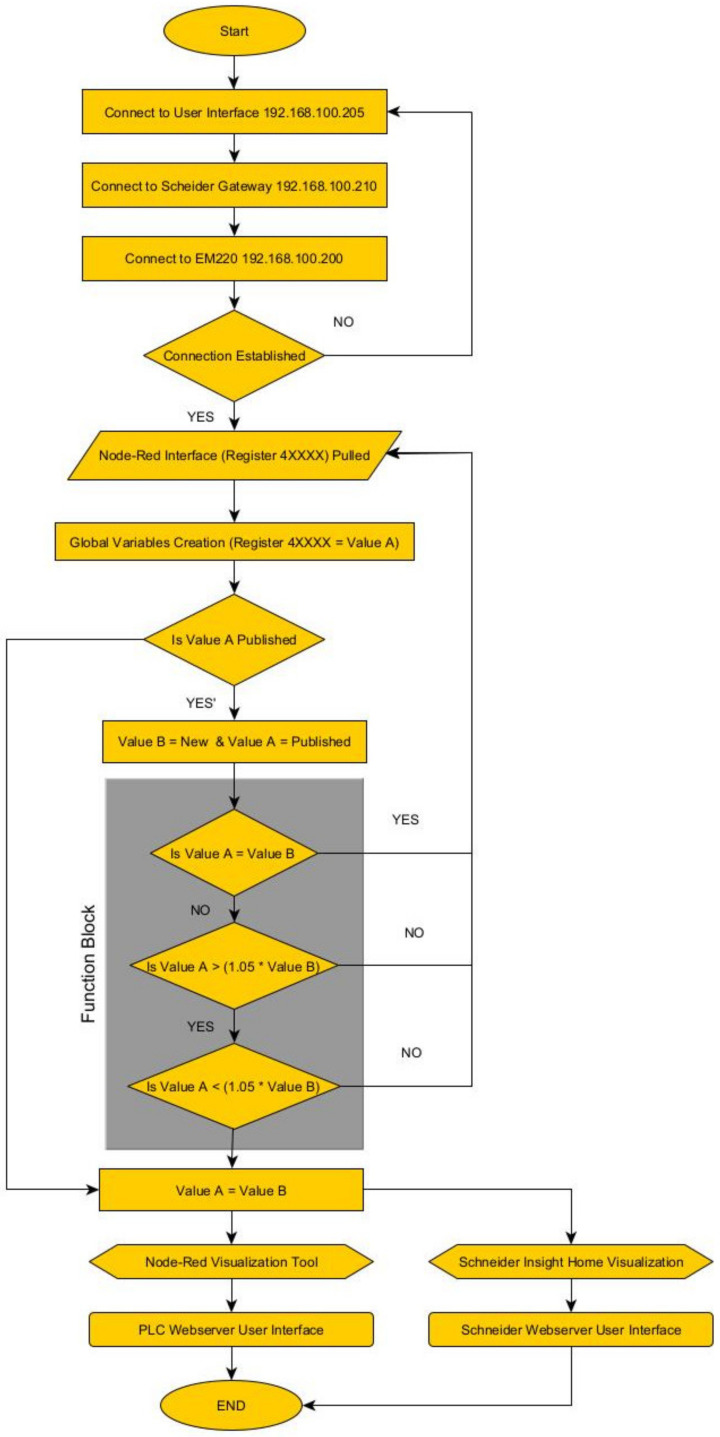
Process flow configured during Application B.

**Figure 15 sensors-24-04116-f015:**
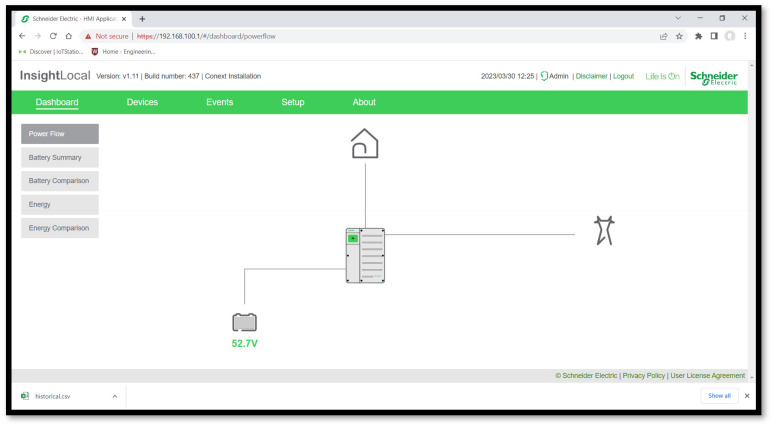
The direct interface from the Schneider gateway, which also illustrates the direct connection to the load, grid, and battery bank.

**Figure 16 sensors-24-04116-f016:**
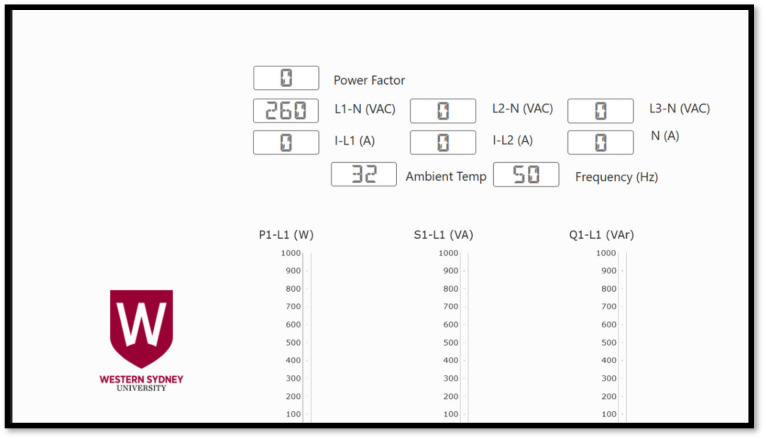
Visualization tool from the UR20, which summarizes the data input from the Schneider gateway and EM220.

**Figure 17 sensors-24-04116-f017:**
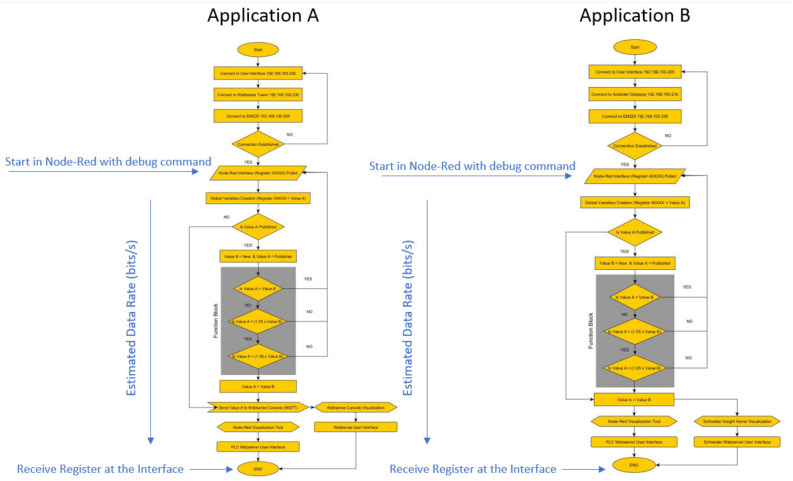
Process flow required to evaluate the latency of the system based on the application required.

**Figure 18 sensors-24-04116-f018:**
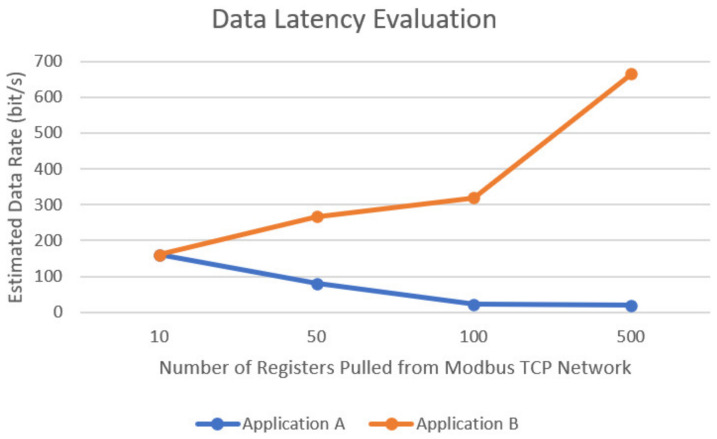
Data latency evaluation between Application A and Application B.

**Table 1 sensors-24-04116-t001:** List of main components utilized in the portable IoT station.

Main Component Description	Manufacturer	Part Number	Quantity
UPS CONTROL MODULE 24VDC 20A/10A	Weidmüller (Detmold, Germany)	1370050010	1
DURA ECO LA-BAT 24V 3.4AH	Weidmüller (Detmold, Germany)	2789900000	1
UR20-WL2000-AC	Weidmüller (Detmold, Germany)	1334950000	1
UR20-16DI-P	Weidmüller (Detmold, Germany)	1315200000	1
UR20-4AI-UI-16	Weidmüller (Detmold, Germany)	1315620000	3
UR20-1COM-232-485-422	Weidmüller (Detmold, Germany)	1315750000	1
UR20-4AI-RTD-DIAG	Weidmüller (Detmold, Germany)	1315700000	1
EM220-RTU-4DI2DO-GW	Weidmüller (Detmold, Germany)	7760051006	1
PRO TOP1 240W 24V 10A	Weidmüller (Detmold, Germany)	2466880000	1
VPU AC II 1+1 R 300/50	Weidmüller (Detmold, Germany)	2591070000	1
ADAPTER USB A RCPT TO USB A RCPT	Weidmüller (Detmold, Germany)	1018840000	2
S202C6 MCB 6kA 2P 6A	ABB (Zürich, Switzerland)	S202C6	1
ABS CASE	Duratech (Auckland, New Zealand)	HB-6387	1
CURRENT CLAMP	FASTRON (Victoria, Australia)	M1100A1A	3
WATTSENSE TOWER	WATTSENSE (Dardilly, France)	TOWER	1
SCHNEIDER INSIGHTHOME	SCHNEIDER (Rueil-Malmaison, France)	865-0330	1

**Table 2 sensors-24-04116-t002:** Modbus TCP address allocation.

Component Type	Description	Modbus TCP Address
Gateway	WATTSENSE TOWER	192.168.100.230
Gateway (865-0330)	SCHNEIDER INSIGHTHOME	192.168.100.210
PLC	UR20-WL2000-AC	192.168.100.220
Energy Meter	EM220-RTU-4DI2DO-GW	192.168.100.200
User Interface (Local)	Computer	192.168.100.205

**Table 3 sensors-24-04116-t003:** Registers collected from EM220 (192.168.100.200).

EM220 (192.168.100.200)			
Register Address	Device Address	Description	Units
30001	0	Phase_1_line_to_neutral_volts	V
30003	2	Phase_2_line_to_neutral_volts	V
30005	4	Phase_3_line_to_neutral_volts	V
30007	6	Phase_1_current	A
30009	8	Phase_2_current	A
30011	10	Phase_3_current	A
30053	52	Total_system_power	W
30063	62	Total_system_power_factor	None
30071	70	Frequency_of_supply_voltages	Hz
30085	84	Total_system_power_demand	W
30201	200	Line_1_to_Line_2_volts	V
30203	202	Line_2_to_Line_3_volts	V
30205	204	Line_3_to_Line_1_volts	V
30235	234	Phase_1_L_N_volts_THD	%
30237	236	Phase_2_L_N_volts_THD	%
30239	238	Phase_3_L_N_volts_THD	%
30241	240	Phase_1_Current_THD	%
30243	242	Phase_2_Current_THD	%
30245	244	Phase_3_Current_THD	%
30403	402	Voltage_2st_63st_Harmonic_L1	%
30527	526	Voltage_2st_63st_Harmonic_L2	%

**Table 4 sensors-24-04116-t004:** Registers collected from UR20-WL2000 (192.168.100.220).

UR20-WL2000-IoT (192.168.100.220)			
Register Address	Device Address	Description	Units
30001	0	Digital Signal 1	I/O
30003	2	Digital Signal 2	I/O
30005	4	Digital Signal 3	I/O
30007	6	Digital Signal 4	I/O
30009	8	Temperature Sensor 1	°C
30011	10	Temperature Sensor 1	°C
30013	12	Temperature Sensor 1	°C
30015	14	Temperature Sensor 1	°C

**Table 5 sensors-24-04116-t005:** Registers collected from Conext XW through InsightHome gateway (192.168.100.210).

InsightHome Gateway (192.168.100.210)			
Register Address	Device Address	Description	Units
40073	73	Inverter-Charger Power Module AC Current Phase A	A
40077	77	Inverter-Charger Power Module Phase Voltage	V
40086	86	Inverter-Charger Power Module Apparent Power	VA
40088	88	Inverter-Charger Power Module Power	W
40090	90	Inverter-Charger Power Module Power Factor	N/A

**Table 6 sensors-24-04116-t006:** Latency results from Application A.

Application A			
Number of Registers	Number of Nodes	Latency (Seconds)	Estimated System Response (bit/s)
10	3	1	160
50	3	10	80
100	3	75	21
500	3	410	20

**Table 7 sensors-24-04116-t007:** Latency results from Application B.

Application B			
Number of Registers	Number of Nodes	Latency (Seconds)	Estimated System Response (bit/s)
10	3	1	160
50	3	3	267
100	3	5	320
500	3	12	667

## Data Availability

All relevant data regarding the design and construction of a portable IoT station are included in this paper.

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
