# Peer review of "Design and Construction of a Portable IoT Station"

_sensors, 2024, doi:10.3390/s24134116_

Round 1
Reviewer 1 Report
Comments and Suggestions for Authors
The abstract provides a concise summary of the study, but it lacks specific details on the results and conclusions drawn from the research.
It should include key findings and the significance of the study.
The introduction effectively sets the context for the research by outlining the importance of IoT systems and the need for synchronised data in various applications. However, it could benefit from a clearer statement of the research problem and objectives. Explicitly stating the research gap the study addresses would strengthen this section.
The literature review is comprehensive, covering relevant previous work and highlighting the need for a portable IoT station. The review could be enhanced by a more critical analysis of the cited works, specifically identifying their limitations and how the current study aims to address them.
The methodology is detailed, describing the hardware and software components used in the portable IoT station. The section on system structure and implementation is well-explained, but the methodology would benefit from a clearer explanation of the experimental setup and procedures used to validate the system. Adding a flowchart or diagram to illustrate the process flow of the system would improve clarity.
System Topology and Implementation The system topology is well-described, with appropriate use of diagrams to illustrate the setup. However, the implementation section could provide more details on the specific configurations and customisations made to the hardware and software components. The section on cybersecurity considerations is crucial but could be expanded to include more details on the specific measures implemented to secure the system.
The results section presents data on system performance, but the analysis is somewhat limited. More detailed statistical analysis and discussion of the results would strengthen this section. For instance, comparing the performance metrics with existing solutions would provide context for the findings.
The discussion on system optimisation is insightful, but it should include specific examples of how the optimisations improved system performance.
The paper effectively demonstrates two applications of the portable IoT station, highlighting its versatility. However, the future work section should provide more concrete plans for further research and development. This could include potential collaborations, specific areas of improvement, and a timeline for future projects.
The conclusions summarise the key findings and contributions of the study but could be more explicit in stating the practical implications and potential impact of the research. The limitations of the study should be acknowledged, and suggestions for future research should be more detailed. References The references are comprehensive and relevant, but ensure all are up-to-date and correctly formatted.
Author Response
We would like to thank you for your thorough review of our manuscript. We appreciate your feedback and suggestions, which have helped us improve the quality of our paper. Please find attached our response.

Reviewer 2 Report
Comments and Suggestions for Authors
The manuscript covers an interesting R&D topic and fits the scope of the Journal. Nonetheless, the paper requires extra efforts to improve its quality and presentation. A set of comments are expounded hereafter.
“PLC” could be added as keyword. On the other hand, “Industrial IoT” is included as keyword but it does not appear within the text of the paper. To solve the latter issue, it is recommended to introduce such a term in the paper.
The contextualization of the work and the Related work section should be enhanced taking into account two aspects. To begin with, the role of PLC is underestimated given its protagonist role in the developed station. This type of equipment has a long trajectory in industrial facilities and is becoming relevant in modern scenarios like the Industry 4.0 and the Industrial IoT. And this leads to the second enhancement, the built station has several features that encompass IoT and industrial components, which is very positive and is aligned with the aforementioned Industry 4.0 and Industrial IoT fields. Both paradigms are attracting a lot of attention from academy and industry and, therefore, the reported station should be framed in both paradigms to reach to a wider audience and to emphasize the relevance of the proposal. Recent publications in this sense should be used to improve the background and framework of the paper. For instance, the following ones could be used:
- Review of Industry 4.0 from the Perspective of Automation and Supervision Systems: Definitions, Architectures and Recent Trends. Electronics 2024, https://doi.org/10.3390/electronics13040782
- Smart Platform for Monitoring and Control of Discrete Event System in Industry 4.0 Concept. Appl. Sci. 2023, 13, 10697. https://doi.org/10.3390/app131910697
- PLC Cybersecurity Test Platform Establishment and Cyberattack Practice. Electronics 2023, 12, 1195. https://doi.org/10.3390/electronics12051195
There is a Modbus secure version released in 2018, a brief mention to this one could be given in order to contextualize the utilization of Modbus TCP with cyber security features.
In the subsection 4.4, “ethernet” is found. The initial letter should be capitalized, as in the rest of the manuscript.
Some abbreviations are not provided, such as LAN or USB.
In figure 5, “Node Red” appears which lacks the hyphen.
The interface RS485 is also used by BMS through the CAN bus, apart from the XanBus. This could be also indicated in an explicit manner.
Modbus TCP and Modbus TCP/IP are both found in the paper. Both manners are correct, but for a coherent presentation, only one of them should be used.
Node-Red is a middleware but this terminology does not appear in the manuscript. Given the increasing use of this middleware, it is suggested to mention this term for an enhanced description.
Another question about Node-Red is that it is unclear for this reviewer where this middleware is executed. This information should be clearly indicated.
Still concerning the Industry 4.0 paradigm, the OPC-UA protocol is considered as the de facto standard for communications in such a paradigm.
What type of battery is used in the HRES? Lead-acid or Lithium-ion technology?
It is suggested to remove the external framework of figure 15 for a better visibility of the chart.
The PLC could be replaced by other model of a different manufacturer?
Even, the title of the paper could be seen as too short and it does not mention the industrial features of the developed IoT station. For example, “based on PLC” or “integrating industrial communications/technologies” could be added in order to be more attractive. Nonetheless, this is a humble suggestion, not a requisite.
Author Response
We would like to thank you for your thorough review of our manuscript. We appreciate your feedback and suggestions, which have helped us improve the quality of our paper. Please find our response attached.

Round 2
Reviewer 1 Report
Comments and Suggestions for Authors
I am happy with the revised version. The authors have addressed all the issues and revised the manuscript quite well. Congratulations!
Reviewer 2 Report
Comments and Suggestions for Authors
The manuscript has been properly revised.